# Modulation of Equid Herpesvirus-1 Replication Dynamics *In Vitro* Using CRISPR/Cas9-Assisted Genome Editing

**DOI:** 10.3390/v16030409

**Published:** 2024-03-06

**Authors:** Rabab T. Hassanien, Côme J. Thieulent, Mariano Carossino, Ganwu Li, Udeni B. R. Balasuriya

**Affiliations:** 1Louisiana Animal Disease Diagnostic Laboratory, School of Veterinary Medicine, Louisiana State University, Baton Rouge, LA 70803, USA; dr_rababtaha@yahoo.com (R.T.H.); cthieulent@lsu.edu (C.J.T.); mcarossino1@lsu.edu (M.C.); 2Department of Pathobiological Sciences, School of Veterinary Medicine, Louisiana State University, Baton Rouge, LA 70803, USA; 3Virology Department, Animal Health Research Institute, Agriculture Research Center (ARC), Dokki, Giza 12618, Egypt; 4Department of Veterinary Diagnostics and Production Animal Medicine, Iowa State University, Ames, IA 50011, USA; liganwu@iastate.edu

**Keywords:** equid alphaherpesvirus-1, EHV-1, CRISPR/Cas9, gene editing, viral replication, antiviral

## Abstract

(1) Background: equid alphaherpesvirus-1 (EHV-1) is a highly contagious viral pathogen prevalent in most horse populations worldwide. Genome-editing technologies such as CRISPR/Cas9 have become powerful tools for precise RNA-guided genome modifications; (2) Methods: we designed single guide RNAs (sgRNA) to target three essential (ORF30, ORF31, and ORF7) and one non-essential (ORF74) EHV-1 genes and determine their effect on viral replication dynamics *in vitro*; (3) Results: we demonstrated that sgRNAs targeting essential lytic genes reduced EHV-1 replication, whereas those targeting ORF74 had a negligible effect. The sgRNAs targeting ORF30 showed the strongest effect on the suppression of EHV-1 replication, with a reduction in viral genomic copy numbers and infectious progeny virus output. Next-generation sequencing identified variants with deletions in the specific cleavage site of selective sgRNAs. Moreover, we evaluated the combination between different sgRNAs and found that the dual combination of sgRNAs targeting ORF30 and ORF7 significantly suppressed viral replication to lower levels compared to the use of a single sgRNA, suggesting a synergic effect; (4) Conclusion: data demonstrate that sgRNA-guided CRISPR/Cas9 can be used to inhibit EHV-1 replication *in vitro*, indicating that this programmable technique can be used to develop a novel, safe, and efficacious therapeutic and prophylactic approach against EHV-1.

## 1. Introduction

Equid alphaherpesvirus-1 (EHV-1) (*Varicellovirus equidalpha 1*) is one of the most common and widespread equine viral pathogens that pose a serious threat to the global equine industry [1,2]. This virus causes respiratory disease, ocular disease (chorioretinopathy), late-term abortions, neonatal mortalities, and neurological disorders (equid herpesviral myeloencephalopathy [EHM]) [3,4,5,6]. After infection, EHV-1 initially replicates in the upper respiratory tract epithelium [3]. Within 12 to 24 h, the virus spreads quickly through the basement membrane to infect the lymphocytes in the lamina propria. By 24 to 48 h, the virus can be detected in the regional lymph nodes of the respiratory tract, where the further replication and infection of leukocytes occurs [1,7,8]. Following viral replication in regional lymph nodes, leukocytes harboring infectious virus are released into the bloodstream, leading to cell-associated viremia in the infected horses [9]. The cell-associated viremia allows the virus to reach sites of secondary infection where contact between infected lymphocytes and the vascular endothelium leads to endothelial cell infection, inflammation, thrombosis, tissue necrosis, and secondary disease manifestations (EHM, abortions, etc.). In pregnant mares, EHV-1 leads to sporadic abortions or abortion storms, and neonatal infections typically lead to fatal neonatal syndrome [5]. EHM is a devastating neurologic form following EHV-1 infection and is associated with infection of the vascular endothelium and subsequent thrombo-ischemic necrosis, mainly within the spinal cord [10]. EHM has been associated with EHV-1 strains possessing a single nucleotide polymorphism (A2254 → G2254) in ORF30 encoding the viral DNA polymerase, resulting in the consequent substitution of asparagine by aspartic acid (N752 → D752) [2,4,11,12]. However, the importance of this mutation for the development of EHM is now debated, and several other host and viral factors are likely associated with the development of this condition in infected horses [2,13,14,15,16]. Like other herpesviruses, EHV-1 can establish latency, during which viral transcription and translation are suppressed, with the exception of the latency-associated transcript (LAT) [17]. Trigeminal ganglia, lymphoid tissues draining the respiratory tract, and peripheral blood leucocytes are recognized as the primary sites of EHV-1 latency [5].

EHV-1 is an enveloped virus belonging to the family *Orthohherpesviridae*, the subfamily *Alphaherpesvirinae*, and the order *Herpesvirales* [18]. It possesses a 150 kbp linear double-stranded DNA genome containing 80 open reading frames (ORFs), encoding 76 different proteins [19]. The gene layout of EHV-1 reveals tightly arranged ORFs with a little intervening sequence, the absence of extensive ORF overlap, and few instances of exon splicing [20]. EHV-1 genes are coordinately expressed and temporally regulated in immediate-early (IE), early (E), and late (L) genes during a productive lytic infection. EHV-1 gene expression begins with the transcription of IE genes. IE proteins are then synthesized in the cytoplasm and migrate to the nucleus to activate the transcription of E genes. The E genes of EHV-1 encode proteins that are essential for viral replication in cultured cells, whilst the L genes encode the viral structural proteins [10,17]. Based on the extensively studied mechanisms of human alphaherpesvirus 1 (herpes simplex virus type 1; HSV-1) replication, it is known that viral DNA replication is initiated once the E proteins are synthesized [21]. The EHV-1 essential genes ORF30, ORF31, and ORF7 are homologs to the HSV-1 genes UL30 (DNA polymerase), UL29 (single strand DNA-binding protein), and UL52 (DNA Helicase/primase subunit), respectively, and have the same DNA replication functions as HSV-1 genes [18,22,23]. EHV-1 ORF74 (US8) is a non-essential gene encoding for the glycoprotein E (gE). The gE promotes cell-to-cell viral spread but is not involved in the process of virus attachment, penetration, maturation, and release [23,24].

During the past decade, there has been a rapid evolution of genome editing technologies in biological research [25,26,27,28,29,30,31,32,33,34], and the clustered regularly interspaced short palindromic repeats (CRISPR)/Cas9 has emerged as one of the most powerful genome editing tools [27,32,35]. The CRISPR/Cas9 system, consisting of a bacterial nuclease (Cas9) and a guide RNA (gRNA) complex, evolved in archaea and bacteria as a form of adaptive immunity to provide a defense mechanism against bacteriophage infection and plasmid transformation [36,37,38,39]. It has been now engineered for the efficient modification of mammalian genomes [40]. The custom-designed single guide RNAs (sgRNAs) recognize their target sequence along the genome and allow Cas9 endonuclease to cleave the DNA double-stranded target sequence at three nucleotides (nt) upstream of the protospacer adjacent motif (PAM) sequence (NGG) [41]. After the Cas9 molecule induces double-stranded DNA breaks, this initiates cellular DNA repair mechanisms as error-prone non-homologous end-joining (NHEJ) and homology-directed repair (HDR) [42]. Random indels (insertions and deletions) occur during NHEJ, leading to mutations and frameshifts in the target DNA, which lead to truncated proteins due to the premature stop codon in a different frame and prevent the translation of the active native protein. Lately, various studies have used the CRISPR/Cas9 technology to impair herpesvirus replication *in vitro* by targeting specific DNA sequences encoding viral proteins, such as HSV-1 [35,43,44,45,46,47,48,49], human alphaherpesvirus 2 (HSV-2) [50,51,52], human betaherpesvirus 5 (cytomegalovirus (CMV)) [35], human gammaherpesvirus 4 (Epstein–Barr virus (EBV) [35,53,54], and bovine alphaherpesvirus type 1 (BHV-1) [55,56].

While various vaccines are currently available for the control/prevention of EHV-1, none of these provide long-term protection against various forms of diseases or prevent the development of EHM [23]. The use of antiviral compounds during EHV-1 disease outbreaks is expensive and not fully effective [2,16,57,58,59,60]. Hence, there is an urgent need for the development of alternative approaches for combatting acute and latent EHV-1 infections in horses. We hypothesized that the CRISPR/Cas9 system could be an effective means of suppressing EHV-1 lytic infection by specifically targeting essential viral genes while targeting non-essential genes, which would not have a significant effect on viral replication. In this study, we performed an *in vitro* screening strategy using designed sgRNAs targeting three essential genes (ORF30, ORF31, and ORF7) and one non-essential gene of EHV-1 (ORF74) and determined their effects on viral replication dynamics *in vitro*. These data indicate the potential use of the CRISPR/Cas9 system as a new therapeutic antiviral strategy for combatting pathogenic equine herpesviruses infections and developing improved prophylactic reagents in the future. This is the first proof-of-concept study that uses CRISPR-Cas9 genome editing on EHV-1, to our knowledge.

## 2. Materials and Methods

### 2.1. Cells and Virus

Equine dermal fibroblasts (E. Derm, NBL-6 ATCC^®^ CCL-57^TM^, Manassas, VA, USA) were maintained in Minimum Essential Medium with Earle’s salts and L-glutamine (EMEM; Corning^®^, Corning, NY, USA), 10% fetal bovine serum (Hyclone^®^, Logan, UT), 1.0 mM sodium pyruvate (Gibco^®^, Carlsbad, CA, USA), 1×X non-essential amino acids (Gibco^®^), penicillin/streptomycin (10,000 U/mL, 10,000 μg/mL; Gibco^®^), and amphotericin B (250 μg/mL, Gibco^®^).

Equine pulmonary artery endothelial cells (EECs, [61]) were maintained in Dulbecco’s MEM (DMEM) with high glucose [4.5 g/L] and without L-glutamine (Corning^®^), supplemented with 10% ferritin-supplemented calf serum, 2 mM L-glutamine (200 mM [100×], Gibco^®^), 1× non-essential amino acids, penicillin/streptomycin, and amphotericin B. Low passage rabbit kidney-13 (RK-13) (passage level 194 to P210) was maintained in EMEM supplemented with 10% ferritin supplemented calf serum (Cytiva, Marlborough, MA, USA), penicillin/streptomycin, and amphotericin B. The cells were incubated at 37 °C and 5% CO_2_ in a humidified incubator.

The EHV-1 T953 strain was used at the third passage on EECs. This strain was isolated from the nasopharyngeal swab of a horse suffering from EHV-1 neurologic disease at the Veterinary Medical Teaching Hospital at The University of Findlay, OH, in 2003 [62] and possesses the A➔G mutation at position 2254 in the viral DNA polymerase gene (ORF30).

### 2.2. Amplification of the Target Genes ORF30, ORF31, ORF7, and ORF74 of EHV-1 T953 Strain

Viral DNA was extracted using tacoTM DNA/RNA Extraction Kit (GeneReach USA, Tauchung, Taiwan) and quantified using Qubit™ dsDNA BR Assay Kits (Invitrogen, Waltham, MA, USA), according to the manufacturer’s instructions. Specific primers for full gene amplification were designed using Geneious software v.6.1.8 (Appendix A; (EHV-1 T953 GenBank Accession number KM593996)). Each target gene sequence was amplified by PCR using the One Step Ahead RT-PCR Kit (Qiagen, Valencia, CA, USA), according to the manufacturer’s instructions, with modifications (i.e., with the removal of the RT step). PCR was performed on a 25 µL reaction mixture containing 10 µL of One Step Ahead RT-PCR master mix (2.5×), 1 µL One Step Ahead RT-mix (25×), 2.5 µL of forward and reverse primers mix (5 µM), 6. 5 µL of RNase-free water, and 5 µL of template DNA. Thermal cycling was performed on a Mastercycler^®^ ×50—PCR Thermocycler (Eppendorf, Hamburg, Germany) using the following conditions: initial step of activation at 95 °C for 5 min, followed by 40 cycles of denaturation at 95 °C for 15 s, annealing at 55 °C for 15 s, and extension at 68 °C for 4 min. Then, final extension took place at 68 °C for 5 min.

The PCR products were analyzed by gel electrophoresis on a 1% agarose gel (Thermo Fisher Scientific, Waltham, MA, USA). Specific amplified DNA fragments of each gene were extracted from gels using the QIAquick^®^ Gel Extraction Kit (Qiagen), according to the manufacturer’s instructions, and quantified using Qubit™ dsDNA BR Assay Kits.

### 2.3. Design, Synthesis, and Preparation of Single Guide RNAs (sgRNAs)

Four single guide RNAs (sgRNAs) specific for the three EHV-1 essential genes (ORF30, ORF31, and ORF7) and one non-essential gene (ORF74) were designed using Benchling (https://www.benchling.com/; accessed on 9 February 2024). sgRNAs with the highest score for specificity and the lowest off-target score within the Equus caballus genome were selected (Table 1). To further ensure the specificity of the selected sgRNAs targets, in silico analysis was performed using the web server Cas-OFFinder (http://www.rgenome.net/cas-offinder/) and the NCBI Basic Local Alignment Search Tool (BLAST; https://blast.ncbi.nlm.nih.gov/Blast.cgi). The selected sgRNAs were synthesized by ThermoFisher Scientific, re-suspended at 100 μM (100 pmol/μL) in 1× TE buffer (pH = 8, 10 mM Tris, 1 mM EDTA; Invitrogen, Waltham, MA, USA) to prepare stock solutions, and stored at −20 °C until use.

### 2.4. In Vitro Cell-Free Cleavage Assay

In order to assess the endonuclease activity of the Cas9 protein and the efficiency of the designed sgRNAs, the PCR product of each gene was incubated with Cas9 protein/sgRNA complexes. The *in vitro* cell-free cleavage assay was performed as follows: a 27 µL reaction mix was prepared containing 3 µL of 300 nM Cas9 protein (Invitrogen) diluted in 1× Diluent B (New England BioLabs, Ipswich, MA, USA), 3 µL of 300 nM sgRNAs, 3 µL of NEBuffer r3.1 10× (New England BioLabs), and 18 µL of nuclease-free water (Invitrogen). After 10 min of incubation at 37 °C, 3 µL of 30 nM PCR product containing the target EHV-1 sequence was added to the mix to reach a 30 µL total volume. The tubes were mixed thoroughly and incubated at 37 °C for 1 h. The reaction was then stopped by adding 1 μL of Proteinase K (New England BioLabs) and incubated at room temperature for 10 min. DNA cleavage was then analyzed by gel electrophoresis in a 1% agarose gel, as described above.

### 2.5. Transfection of E. Derm Cells with Cas9 Protein

A total of 1 × 10^5^ E. Derm cells in 500 µL complete EMEM were seeded in 24-well plates containing 12 mm circle micro cover glass (VWR, Radnor, PA) and incubated at 37 °C and 5% CO_2_ for 24 h. The next day, the transfection reaction was prepared as follows: tube A was prepared with 25 µL of Opti-MEM (Gibco^®^), 2 µg (12 pmol) of Cas9 protein, and 4 µL of Cas9 plus reagent (Invitrogen). The tube A was mixed well and incubated at room temperature for 5 min. Tube B was prepared with 25 µL of Opti-MEM and 1.5 µL of lipofectamine CRISPRMAX reagent (Invitrogen). The solution in tube A was immediately added to that in tube B and mixed well, and the complex was incubated for 10 min at room temperature. During this time, the media on the cells were replaced with 500 µL of Opti-MEM medium (reduced serum media) or complete EMEM (10% fetal bovine serum), and 50 µL of transfection reaction was added to the cells. The cells were incubated at 37 °C for 12, 24, 48, and 72 h post-transfection (hpt) and then analyzed by an indirect immunofluorescence assay (IFA).

### 2.6. Immunofluorescence Assay (IFA) for the Evaluation of Cas9 Protein Transfection Efficiency in E. Derm Cells

The cell culture medium was removed, and the cells were rinsed three times with 500 µL of 1× PBS containing 10 mM of glycine (Alfa Aesar, Heysham, UK). The cells were subsequently fixed with 500 µL of 4% paraformaldehyde (pH = 7.4) for 1 h at room temperature and washed three times, as previously described. The cells were then permeabilized by adding 400 µL of 0.1% Triton X-100 (Sigma-Aldrich, St. Louis, MO, USA) for 15 min at room temperature. The cells were washed three times, and 500 µL of blocking solution (5% fetal bovine serum in 1× PBS) was added and incubated at room temperature for 30 min. Then, the cells were stained with 200 µL of an anti-Cas9 mouse monoclonal antibody (7A9-3A3, Invitrogen) diluted at 5 µg/mL in blocking solution for 1 h at room temperature. After washing, 200 µL of diluted fluorescent dye-labeled secondary antibodies (goat anti-mouse, Alexa Fluor 488, Invitrogen) diluted at 1 µg/mL in 1× PBS along with the compatible counterstain for the actin filament Texas Phalloidin (Invitrogen) (0.165 µM) was added, and the cells were incubated for 45 min at room temperature in the dark. Finally, 22 µL of Hoechst 33342 solution (20 µg/mL, Invitrogen) was added to each well and incubated for an additional 15 min. After washing, the coverslip was mounted on a glass slide using Fluromount-G^TM^ mounting medium (SouthernBiotech, Birmingham, AL, USA), and the cells were examined using fluorescence microscopy (Leica DM IRB Heerbrugg, Switzerland). The number of Cas9 dots per cell was calculated from two fields of three different experiments.

### 2.7. Transfection of Cas9 Protein and sgRNAs Specifically Targeting ORF30, 31, 7, 74 on EHV-1-Infected E. Derm Cells

A total of 1 × 10^5^ E. Derm cells in 500 µL complete EMEM were seeded in 24-well plates and incubated at 37 °C for 24 h. The following day, EHV-1 T953 strain was added to the cells at a multiplicity of infection (MOI) of 0.05 and incubated for 1 h at 37 ^°C^ with rocking every 10 min. The inoculum was removed, the cells were washed three times with EMEM containing 2% fetal bovine serum, and 500 µL of Opti-MEM was finally added to each well. During the adsorption period, the transfection reaction of each sgRNA/Cas9 combination was prepared as previously described; tube A was prepared with 25 µL of Opti-MEM, 2 µg (12 pmol) of Cas9 protein, 7.4 µM (12 pmol) of sgRNA, and 4 µL of Cas9 plus reagent. Tube B was prepared with 25 µL of Opti-MEM and 1.5 µL of lipofectamine CRISPRMAX reagent. A transfection reaction containing Cas9 only and each sgRNA only was included in the experiment as controls. At 72 hpt, the supernatant was collected, and the plates were subjected to two cycles of freeze–thaw. Cellular debris was removed by centrifugation at 4000× *g* for 5 min, and the cell lysates were stored at −80 °C until they were used.

### 2.8. Virus Titration by Plaque Assay

Cell culture supernatants were subjected to ten-fold serial dilutions in EMEM and inoculated in RK-13 cells pre-seeded in 12-well plates. After 1 h of adsorption at 37 °C, with rocking every 10 min, the cells were overlaid with 2 mL of a 1:1 ratio of complete Earl’s balanced salt solution media containing 1% agarose (EBSS, Gibco^®^) with complete EMEM and incubated for 48 h at 37 °C. Then, 1 mL of a second overlay containing a 1:1 ratio of complete EBSS containing 1% agarose and 0.01% Neutral red (Sigma Millipore, St. Louis, MO, USA) and complete EMEM was added, and the plates were incubated at 37 °C for an additional 24 h. The plaques were counted after 72 h post-infection (hpi), and the viral titers were determined as plaque forming units per ml (PFU/mL).

### 2.9. Viral DNA Extraction and Quantification

Viral DNA was extracted from the cell supernatant 72 hpi using the taco^TM^ DNA/RNA extraction kit, according to the manufacturer’s instructions, and stored at −80 °C until use. Real-time quantitative PCR (qPCR) targeting ORF30 was performed on a 25 µL reaction mixture containing 12.5 µL of QuantiTech Mutiplex PCR Master Mix (Qiagen), 1.25 µL of primers (8 µM) (Forward: 5′ TCTGGCCGGGCTTCAAC 3′; Reverse: 5′TTTGGTCACCCACCTCGAA 3′) and probe (3.5 µM) ([6-FAM] ATCCGTCGACTACTCG [BHQ1a-6FAM]) mix [63], 6.25 µL of nuclease-free water, and 5 µL of template DNA. Thermal cycling was performed on a 7500 Fast Real Time PCR System (Life Technologies, Carlsbad, CA, USA) in standard mode using the following conditions: an initial step of denaturation at 95 °C for 15 min, followed by 40 cycles of amplification at 95 °C for 15 s and annealing/extension at 58 °C for 1 min. EHV-1 DNA quantification was performed using a serial dilution (10^8^ copies/mL to 10^3^ copies/mL) of pEHV1G plasmid containing a 165 bp insert sequence located at positions 1306 to 1470 of the EHV-1 ORF30 gene (KM593996.1). The insert sequence was as follows: 5′TCGCATGGCCAGCCAGTCGCGCAGCAAGATGCCAAGCAGGCTTTCGCGAATATGGGCGTGGACAAAAAATAACTTTTGGTCACCCACCTCGAACGTCGAGTAGTCGACGGATGGTTGAAGCCCGGCCAGATCCACTTCATCGAGCGCCAGGGTGGTGAAACAGAG 3′.

### 2.10. Genomic Cleavage Assay

The viral genomic modification efficiency induced by the CRISPR/Cas9 system was determined based on the total cell lysate DNA using the GeneArt cleavage detection kit (Invitrogen), following the manufacturer’s instructions. Briefly, PCR amplicons were obtained by amplification using specific primer sequences designed to target the cleavage site for each sgRNA (Appendix A). A 50 µL reaction was prepared as follows; 18 µL of nuclease-free water, 25 µL AmpliTaq Gold^®^ 360 Master Mix (Applied Biosystems, Whaltham, MA, USA), 2 µL of forward and reverse primer mix (5 µM), and 5 µL of template DNA. The thermal profile was performed using a Mastercycler^®^ ×50—PCR Thermocycler, following the manufacturer’s instructions: enzyme activation at 95 °C for 10 min, followed by 40 cycles of denaturation at 95 °C for 30 s, annealing at 60 °C for 30 s, extension at 72 °C for 36 s, final extension at 72 °C for 7 min, and hold at 4 °C. Then, 2 µL of amplified PCR products was mixed with 1 µL 10× detection reaction buffer and 6 µL of nuclease-free water, denatured by heating and subsequent annealing to form heteroduplex DNA (95 °C for 5 min, 95 °C ➔ 85 °C (−2 °C/s), 85 °C ➔ 25 °C (−0.1 °C/s, hold at 4 °C)). The reaction was then digested with 1µL of detection endonuclease enzyme for 1 h at 37 °C. The PCR products were analyzed using 2% agarose gel electrophoresis, as described above.

### 2.11. Sequencing of the CRISPR/Cas9 Edited Viruses

The sequencing of the PCR products of CRISPR-targeted EHV-1 genes was performed to assess for editing at the target sites for selected sgRNAs (sgRNA ORF30_3, sgRNA ORF31_2, and sgRNA ORF7_3). The genomic target sites of each respective sgRNAs were amplified using the DNA extracted from the cell lysate, as mentioned above, and the specific primer sequences designed (Appendix A). Amplification was performed by PCR using the One Step Ahead RT-PCR Kit (Qiagen), according to the manufacturer’s instructions, as previously described in paragraph 2.2 in Materials and Methods. The PCR products were analyzed by gel electrophoresis using a 2% agarose gel (Thermo Fisher Scientific). Amplified DNA fragments of each sgRNA were extracted from gels using the QIAquick^®^ Gel Extraction Kit (Qiagen), according to the manufacturer’s instructions, quantified using Qubit™ dsDNA BR Assay Kits, and sent for next-generation sequencing (NGS) and bioinformatic analysis to the Veterinary Diagnostic Laboratory, Iowa State University. The sequencing library was prepared using the Nextera XT DNA library preparation kit (Illumina, San Diego, CA, USA) with dual indexing. The pooled libraries were sequenced on an Illumina MiSeq platform using the 300-Cycle v2 Reagent Kit (Illumina) by following the standard Illumina instructions. The sequences of PCR products were assembled as described previously [64,65]. Briefly, publicly available whole genome sequences of the EHV-1 were downloaded from NCBI to serve as a collection of reference sequences. Quality-trimmed reads were mapped against the reference sequences by using BWA-MEM [66]. Mapped reads were extracted by SAMtools and seqtk (https://github.com/lh3/seqtk) [67]. De novo assembly was performed using ABySS v1.3.9 [68]. The resulting sequences were manually checked and trimmed in SeqMan Pro 15 (DNASTAR^®^ Lasergene 15 Core Suite). An in-house script was used to extract all the insertions/deletions and count the frequency of each insertion and deletion.

### 2.12. Cell Viability Assay

A total of 2 × 10^4^ E. Derm cells in 100 µL media were seeded in white opaque 96-well plates (Greiner Bio-One, Kremsmunster, Austria) and incubated at 37 °C for 24 h. Then, the medium was removed, the cells were washed with EMEM containing 2% FBS, and 100 µL of Opti-MEM or EMEM was added to the specific wells. E. Derm cells were transfected with 10 µL of the transfection complex containing sgRNA/Cas9, Cas9 only, and sgRNA only and incubated for 72 h at 37 °C. Cell viability was measured by ATP measurement using the CellTiter Glo^®^ Luminescent Cell Viability Kit (CTG; Promega, WI, USA), according to the manufacturer’s instructions. The luminescence signal was acquired using a Spectra max M2^®^ luminometer (Sunnyvale, CA, USA), and the percentage of viability was calculated using the following formula: Viability (%) = ([RLU_T_ − RLU_M_] × 100) ÷ (RLU_C_ − RLU_M_), where RLU_T_ corresponds to the relative luminescence of the treated cell, RLU_M_ corresponds to the relative luminescence of the medium, and RLU_C_ corresponds to the relative luminescence of the control cells.

### 2.13. Statistical Analysis

Statistical analyses were performed using GraphPad Prism v9.3.1 statistical analysis software (GraphPad, San Diego, CA, USA). Specific statistical tests are mentioned in the respective figure legends. Data are expressed as the mean ± standard deviation (SD) and were considered significant at a *p*-value ≤ 0.05.

## 3. Results

### 3.1. The Selected sgRNAs Are Specific for the Targeted EHV-1 Genes

Four EHV-1 genes were selected as targets for sgRNA design: ORF30, encoding for the DNA polymerase; ORF31, encoding for a major DNA-binding protein; ORF7, encoding for a helicase-primase subunit; and ORF74, encoding for glycoprotein E (gE; non-essential). The selected EHV-1 genes were first amplified by PCR using specific primers. A specific band at the expected molecular weight was observed at 3919 bp, 3823 bp, 3600 bp, and 1854 bp for ORF30, ORF31, ORF7, and ORF74, respectively (Appendix A). The *in vitro* cell-free cleavage assay was then performed by incubating each DNA amplicon with each specific sgRNA in the presence or absence of the Cas9 protein, and the results were analyzed by gel electrophoresis. For all the PCR products incubated in the presence of both sgRNAs and the Cas9 protein, three specific bands were observed; one parental band of a high molecular weight and two additional bands of lower molecular weights associated with DNA cleavage (Figure 1). Only the parental band was observed for the PCR products treated with the Cas9 protein only or with the sgRNAs only, showing the absence of cleavage. These results indicated that all the designed sgRNAs, in the presence of the Cas9 protein, have the ability to cleave the EHV-1 genome at the expected sites.

### 3.2. Opti-MEM Increases the Transfection Efficiency of Cas9 in E. Derm Cells

In order to evaluate the efficiency of the Cas9 protein transfection in E. Derm cells, two different media were tested: EMEM + 10% FBS and Opti-MEM. The cells were incubated for 12, 24, 48, and 72 hpt, and IFA was performed to assess the Cas9 uptake level (Figure 2A). The Opti-MEM medium significantly increased the uptake of Cas9 in E. Derm cells by >30-fold from 12 hpt compared to the EMEM + 10% FBS, while EMEM + 10% FBS decreased Cas9 protein expression as a function of time (*p* ≤ 0.001; Figure 2B). Even though the Opti-MEM significantly increases the uptake of Cas9 in E. Derm cells, it concurrently reduces cellular viability by 31.8 ± 4.5% (*p* ≤ 0.001) at 72 hpt compared to EMEM + 10% FBS (Figure 2C). Based on the significant increase in the uptake of the Cas9 protein in E. Derm cells using Opti-MEM, this medium was used in subsequent experiments.

### 3.3. CRISPR/Cas9 Complexes Targeting Essential Genes Inhibit EHV-1 Replication In Vitro

In order to evaluate if the selected sgRNAs can inhibit the replication of EHV-1 *in vitro*, E. Derm cells were infected with the EHV-1 T953 strain at an MOI of 0.05 and treated immediately after with the mix of the Cas9 protein and specific sgRNAs. The supernatant was collected at 72 hpt for viral genomic copy quantification (i.e., qPCR) and virus titration (i.e., plaque assay). The four sgRNAs targeting ORF30 significantly reduced the viral genome copies/mL in the tissue culture supernatant (ORF30_1: *p* ≤ 0.001; ORF30_2: *p* = 0.028; ORF30_3: *p* ≤ 0.001; ORF30_4: *p* ≤ 0.001) (Figure 3A). Viral titers were significantly reduced in the presence of sgRNAs ORF30_1 (*p* = 0.042), ORF30_3 (*p* ≤ 0.001), and ORF30_4 (*p* = 0.005) but not with sgRNA ORF30_2 (*p* = 0.101) (Figure 3B). With a reduction of 2.21 ± 0.27 log_10_ viral copies/mL and 2.19 ± 0.53 log_10_ PFU/mL, sgRNA ORF30_3 was the most effective in reducing EHV-1 replication. In regard to the sgRNAs targeting ORF31, only the sgRNA ORF31_2 showed a significant effect, with a reduction of 1.04 ± 0.26 log_10_ viral copies/mL (*p* = 0.003; Figure 3C) and 0.98 ± 0.17 log_10_ PFU/mL (*p* ≤ 0.001; Figure 3D). Three of the four sgRNAs targeting the ORF7 significantly reduced viral genome copy numbers (Figure 3E). Among them, ORF7_3 was the most effective, with a reduction of 1.45 ± 0.04 log_10_ genomic copies/mL (*p* ≤ 0.001). ORF7_1 (*p =* 0.017) and ORF7_2 (*p* = 0.005) were less effective, with a reduction of < 1 log_10_/mL, while ORF7_4 did not show any effect (*p* = 0.222). With a reduction of 1.77 ± 0.23 log_10_ PFU/mL (*p* > 0.001), only the sgRNA ORF7_3 significantly reduced the production of infectious viral particles (Figure 3F). Among the four sgRNAs targeting ORF74, only sgRNA ORF74_2 significantly reduced the number of EHV-1 genomic copies/mL of the tissues culture supernatant (0.77 ± 0.18 log_10_/mL, *p* = 0.045; Figure 3G), but none of them were effective in reducing viral titers (Figure 3H). These results demonstrate that using the CRISPR/Cas9 technique with sgRNAs targeting the three EHV-1 essential genes impaired virus replication, while almost no effect was observed when targeting the non-essential gene ORF74. The most effective sgRNAs targeting each essential gene (ORF30_3, ORF31_2, and ORF7_3) were selected for further analysis.

The cytotoxicity of the Cas9 protein and of the sgRNAs ORF30_3, ORF31_2, and ORF7_3 was evaluated on E. Derm cells at 72 hpt by ATP measurement using the CellTiter-Glo^®^ Luminescence Cell Viability Assay (Figure 4). When compared to the control cells cultured in Opti-MEM medium, the Cas9 protein and all the sgRNAs did not significantly reduce the viability of the transfected cells.

### 3.4. Essential Gene Targeting via sgRNA CRISPR/Cas9 Has a Dose-Response Effect on EHV-1 Replication Dynamics

A dose-response assay was performed with three different concentrations (2.5 µM, 3.7 µM, and 7.4 µM) of sgRNA ORF30_3, ORF31_2, and ORF7_3, and their effect on EHV-1 replication was analyzed, as described above. The results showed that these three sgRNAs reduced viral genomic copies/mL in a dose-dependent manner (ORF30_3: *p* = 0.006, Figure 5A; ORF31_2: *p* = 0.024, Figure 5B; ORF7_3: *p* = 0.036, Figure 5C), confirming the antiviral effect of these sgRNAs against EHV-1.

### 3.5. sgRNA-Targeted Editing Mainly Induces Frameshift Deletions in EHV-1-Specific ORFs with Predicted Premature Termination of Protein Translation

EHV-1 editing sites containing insertions, deletions, or mutations (indels) were detected using the GeneArt cleavage detection kit. In the cell lysate of the three selected sgRNAs targeting essential genes (ORF30_3, ORF31_2, ORF7_3) and one sgRNA targeting a non-essential gene (ORF74_2), cleavage detection was observed with the presence of three amplicons (parental band and two cleaved bands) in each examined sample, whereas only one intense parental band was observed for each specific control (Figure 6A). These findings indicate the presence of both wild-type and cleaved EHV-1 genome in the examined samples and predict editing of the EHV-1 genome at the targeted sites. To confirm these observations, viral DNA sequencing was subsequently performed by NGS. The results showed that the sgRNA ORF30_3 induced a seven-nucleotides deletion (AACTCAA; 97,437–97,443) variant with 9.07% frequency compared to the wild-type strain (Figure 6B). The presence of variants with two nucleotide deletions (CG; 58,221–58,222 and GA; 141,755–141,756) at frequency rates of 5.94% and 2.65% was observed in the presence of the sgRNAs ORF31_2 and ORF7_3, respectively (Figure 6C,D). These are all predicted to be frameshift deletions inducing occurrence of a STOP codon and premature protein translation termination (Figure 6). These results confirmed that the reduction in the virus produced is induced by the effect of our CRIPR/Cas9 assay.

### 3.6. Combined ORF30- and ORF7-Targeting sgRNA has a Synergistic Effect on the Inhibition of EHV-1 Replication

In order to determine whether the combination of various sgRNAs targeting different EHV-1 genes has a synergistic effect in inhibiting viral replication, the selected sgRNAs targeting ORF30_3, ORF31_2, and ORF7_3 were evaluated in pairwise and triplex combinations, as previously described. To compare the results of combinations with the effect of a single sgRNA treatment, the final total concentration of sgRNAs is always equal to 7.4 µM. As shown previously, sgRNAs ORF30_3, ORF31_2, and ORF7_3 administered individually caused a significant reduction in EHV-1 genomic copies and infectious titers compared to the infected control cells (Figure 7A,B). The combination of the sgRNAs ORF30_3 + ORF7_3 was the most effective combination and significantly reduced the EHV-1 genomic copies and infectious viral particles produced by 3 ± 0.07 log_10_ viral copies/mL (*p ≤* 0.001) and 3.22 ± 0.27 log_10_ PFU/mL (*p* ≤ 0.001), respectively. When compared to ORF30_3 alone, this combination significantly reduced the EHV-1 genomic copies and infectious viral particles produced by 0.82 ± 32 log_10_ viral copies/mL (*p* ≤ 0.001) and 1.03 ± 0.68 log_10_ PFU/mL (*p* = 0.003), respectively. Additionally, the combination showed a significant reduction in the EHV-1 genomic copies and infectious viral particles produced by 1.58 ± 0.05 log_10_ viral copies/mL (*p* ≤ 0.001) and 1.46 ± 0.19 log_10_ PFU/mL (*p* = 0.006) when compared to ORF7_3 alone, suggesting a synergistic effect. The combination of the sgRNAs ORF30_3 + ORF31_2 reduced the production of EHV-1 by 1.31 ± 0.30 log_10_ viral copies/mL (*p* ≤ 0.001) and 1.42 ± 0.13 log_10_ PFU/mL (*p* ≤ 0.001). However, this combination is significantly less effective than ORF30_3 alone in reducing the production of viral genomic copies (*p* = 0.002) and infectious particles (*p* = 0.004). Referring to the combination of sgRNAs ORF31_2 + ORF7_3, it showed the lowest effect in inhibiting EHV-1 replication, with a reduction of 0.95 ± 0.44 log_10_ viral copies/mL (*p* = 0.001) and 1.24 ± 0.36 log_10_ PFU/mL (*p* ≤ 0.001). When these sgRNAs were used as a triplex combination, they significantly reduced the EHV-1 copies number and infectious particles by 1.82 ± 0.30 log_10_ (*p* ≤ 0.001) and 2.00 ± 0.23 log_10_ (*p* ≤ 0.001), respectively. However, the triplex combination is less effective in reducing the number of EHV-1 copies (*p* ≤ 0.001) and infectious particles (*p* ≤ 0.001) than the pairwise ORF30_3 + ORF7_3. No cytotoxicity on E. Derm cells was observed at 72 hpt for all the different combinations tested here (Figure 4). In conclusion, the combination of sgRNAs ORF30_3 + ORF7_3 showed the most pronounced synergistic effect on EHV-1 replication *in vitro*.

## 4. Discussion

EHV-1 is one of the most common viral pathogens circulating in the horse population and is associated with different disease outcomes including respiratory disease, late-term abortions and still-births, neonatal deaths, chorioretinopathy, and neurological disease [3]. During the last decade, several approaches have been taken to develop different vaccines and antiviral compounds for the prevention and control of EHV-1 outbreaks around the world [23,59]. However, none of these are fully effective against EHV-1. The CRISPR/Cas9 system has been presented as a potent genome-engineering tool for a variety of applications to be used by researchers, including gene therapy, genome editing, genetic screening, and RNA display [40,69]. Furthermore, this technology has been proven to be effective in inhibiting DNA virus replication *in vitro* [35,48]. Here, we performed a proof-of-concept study to show that the CRISPR/Cas9 genome engineering system can be efficiently used to inhibit EHV-1 replication (i.e., the lytic cycle) *in vitro*. For this purpose, sgRNAs targeting three EHV-1 essential genes and one non-essential gene were designed in silico. ORF30, ORF31, and ORF7 were chosen as CRISPR/Cas9 targets because these genes have a major role in EHV-1 lytic life cycle, and knocking them out may affect viral infection. ORF74 is not essential for lytic virus replication and was selected as a control.

Designing sgRNAs targeting EHV-1 genome was challenging because commonly used software programs for custom designing sgRNAs, such as E-CRISP, CRISPOR, Deskgen, and CHOPCHOP, do not have the EHV-1 genome in their databases. For this reason, we used Benchling software to design the sgRNAs as a starting point in our CRISPR experiment, as it allows for the upload of EHV-1 target genes. We selected the four sgRNAs with the highest specificity and the lowest off-target score within the *Equus caballus* genome, which are important factors to consider. The *in vitro* cell-free cleavage assay directly on viral DNA is a sensitive, rapid, and excellent preliminary tool for the validation of the sgRNAs [43,47,69]. Here, this technique allowed us to confirm that 100% of the designed sgRNAs can target and induce a specific cleavage of EHV-1 DNA by the Cas9 protein.

Our results demonstrated that some sgRNAs targeting EHV-1 essential genes could considerably inhibit EHV-1 replication in E. Derm cells (e.g., sgRNA ORF30_3, ORF31_2, and ORF7_3). Although they demonstrated effective cleavage in the *in vitro* cell-free cleavage experiment, other sgRNAs (e.g., ORF31_1, ORF31 _3, ORF31 _4, and ORF7 _4) did not exhibit any effect on EHV-1 replication in E. Derm cells. The sgRNA sequence is the key for successful genome editing, and different sgRNAs sequences targeting the same gene might have a significant variation in their efficacy for CRISPR/Cas9 editing [35]. The targeted allele’s PAM sequence, downstream nucleotides, and sequence characteristics of the sgRNA together play a role in the potency of sgRNAs. Our findings are consistent with several previous reports, in which the authors targeted different essential genes to inhibit the replication of herpesviruses [35,70,71,72].

Comparing the effect of all the sgRNAs tested in this study, ORF30_3 is consistently more efficient than other sgRNAs in inhibiting viral replication. Our results are consistent with a previous experiment conducted on HSV-1, in which the authors reported that sgRNAs targeting the UL30 (ORF30) of HSV-1 were the most effective in reducing viral replication and output [47]. Regarding the ORF31 and ORF7 of EHV-1, we determined that ORF7_3 and ORF31_2 showed the most significant reduction for EHV-1 growth in the E. Derm cells, the former being more effective than the latter. These results came in accordance with another study conducted on HSV-1 which indicated that targeting UL52 (ORF7) was more effective in inhibiting HSV-1 replication than UL29 (ORF31) [35].

In this study, we have decided to use purified Cas9 protein and sgRNA instead of plasmid-based CRISPR/Cas9 system. Indeed, this approach has been shown to have a quick action, efficient gene editing, low off-target effects, and toxicity [73,74,75,76]. However, the performance is lower when compared to that of the plasmid-based CRISPR/Cas9 system. Our data clearly showed that using the purified Cas9 protein and sgRNA is an efficient way to use the CRISPR/Cas9 system in E. Derm cells. Furthermore, this approach would be more amenable for future in vivo applications, as the purified Cas9 protein could represent a safer and more efficient method for targeting animal cells during therapeutic applications when compared to the plasmid-based CRISPR/Cas9 system, which may increase the risks of DNA plasmid integration into the host genome and induce non-specific gene editing in the virus or host genome [40,44,48,69,75,77,78,79,80,81]. Nevertheless, our study has shown a lower impact on viral titer reduction when utilizing the Cas9 protein to target EHV-1 ORF30, ORF7, and ORF31 compared to other studies that have utilized a plasmid-based CRISPR/Cas9 system against the corresponding HVS-1 essential genes [35,47,48]. Although direct comparisons between our results and those previously published cannot be made, we hypothesize that using a plasmid-based CRISPR/Cas9 system that encodes the Cas9 protein and sgRNA induces a stronger effect on suppressing virus replication due to the constant expression of sgRNAs and the Cas9 protein. Thus, this technique should be considered in future *in vitro* studies.

It was previously reported that an EHV-1 mutant with deletions in ORF74 (gE) has a comparable replication growth to the parental strain *in vitro* [82]. Targeting ORF74 in the present study did not affect, or had a low effect on, viral replication dynamics, although the cleavage of the viral genome was confirmed by the cleavage detection kit. This result is in accordance with a recent study conducted on BHV-1, where the gE gene was used as a control to evaluate the non-specific effects of CRISPR components on virus replication [83].

Finally, we evaluated dual sgRNAs targeting two different essential genes and found a potential synergistic effect of the combination sgRNAs ORF30_3 + ORF7_3 in reducing viral replication. This indicated that targeting multiple essential EHV-1 genes concomitantly significantly increased the inhibition of viral replication compared to targeting only one single gene, which was previously reported against human herpesviruses [35,44,47,70]. We also tried a combination of three sgRNAs (ORF30_3 + ORF31_2 + ORF7_3) targeting different essential genes. Although this triple-combination reduced the EHV-1 replication, the effect was lower than the combination of the sgRNAs ORF30_3 + ORF7_3. This can be explained by the usage of ORF31_2, which gave a less effective reduction in EHV-1. To our knowledge, this was the first experiment where triple combinations of three sgRNAs were used in CRISPR to study the effects on herpesvirus replication.

Analyzing the next-generation sequence results revealed that the non-homologous end-joining cell-repairing mechanism for the CRISPR/Cas9-induced dsDNA breaks resulted in mutant viruses with frameshift deletions, inducing the occurrence of a STOP codon and premature protein translation termination. Hence, these finding may impact the biology of virus replication and the production of the targeted proteins [35,47]. Therefore, in this study, we suggest using our designed CRISPR/Cas9 with the combination sgRNAs ORF30_3 + ORF7_3 platform as a targeted and efficient tool for developing antiviral agents and potential future therapeutic applications for EHV-1.

## 5. Conclusions

We have shown that CRISPR/Cas9 technology can effectively modulate EHV-1 infection *in vitro*. Furthermore, by combining two sgRNAs targeting two different essential EHV-1 genes (sgRNAs ORF30_3 + ORF7_3), we efficiently increased the virus replication inhibition in E. Derm cells. These new insights may allow for the design of effective therapeutic strategies for targeting equine and other animal herpesviruses during both productive and latent infections in the future. The efficiency of this system may be further improved by using a vector-based CRISPR/Cas9 system to abolish the virus replication completely.

## Figures and Tables

**Figure 1 viruses-16-00409-f001:**
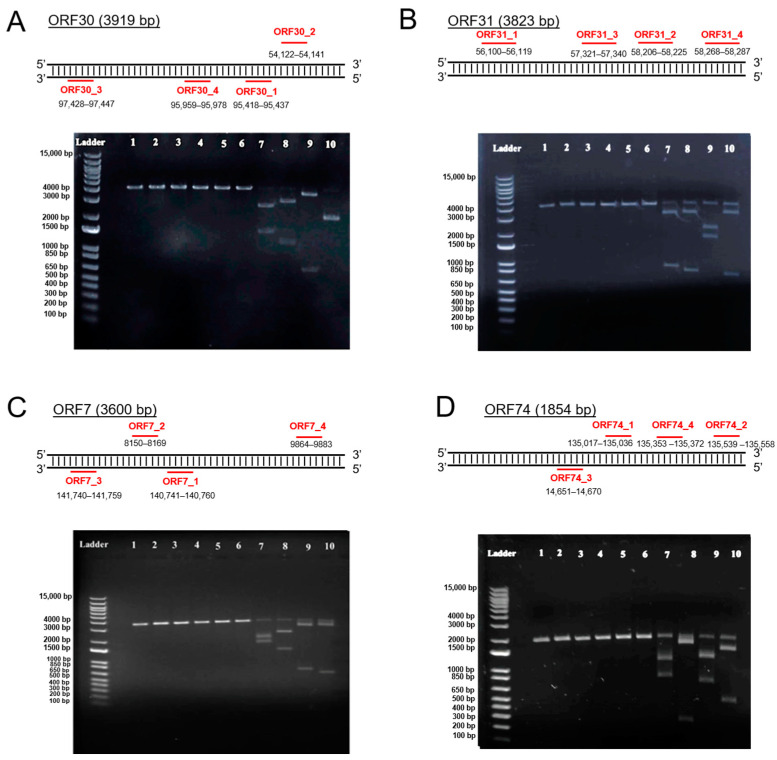
*In vitro* cell-free cleavage assay used to assess the endonuclease activity of the Cas9 protein and the efficiency of the designed sgRNAs. Amplified DNA from targeted genes in the EHV-1 genome was incubated in the presence or absence of the Cas9 protein and specific sgRNAs and analyzed by gel electrophoresis. A schematic representation of the target region of each sgRNA is illustrated for each gene: sgRNAs for ORF30 (**A**), sgRNAs for ORF31 (**B**), sgRNAs for ORF7 (**C**), and sgRNAs for ORF74 (**D**). Lane (1): Control PCR product for the target gene; lane (2): PCR product + Cas9 protein; lanes (3–6): PCR product + sgRNA_1 to sgRNA_4; lane (7–10): PCR product + Cas9 protein + sgRNA_1 to sgRNA_4. Only two bands were observed when EHV-1 ORF30 PCR product was treated with the Cas9 protein and sgRNA ORF30_4 ((**A**), lane 10) because sgRNA ORF30_4 was designed to target positions 95,959–95,978 bp, centrally located within the ORF30 amplicon; this yielded two cleaved bands of similar molecular weights.

**Figure 2 viruses-16-00409-f002:**
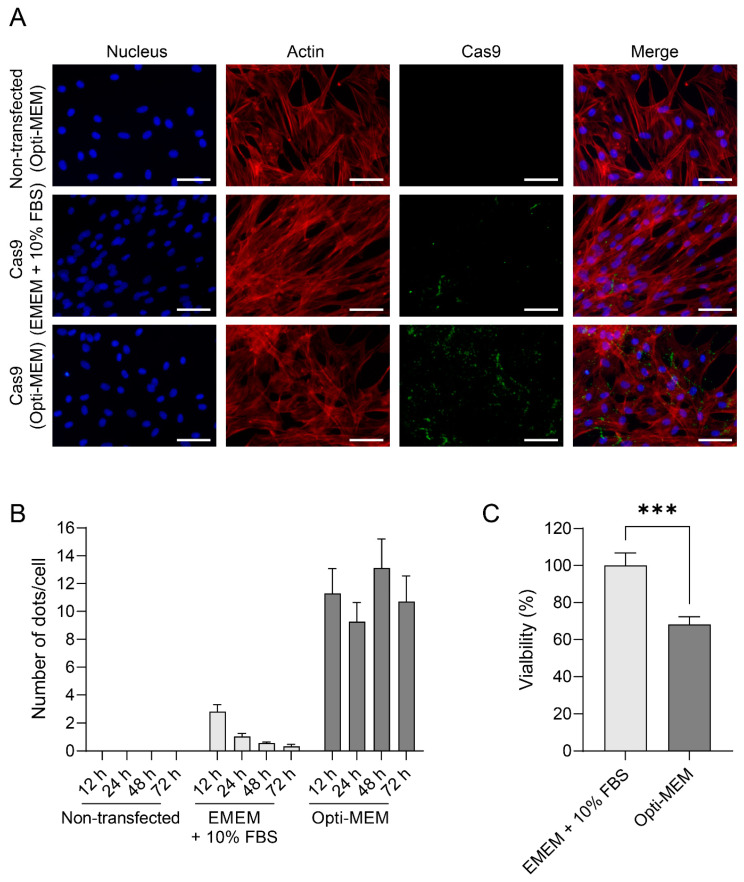
Evaluation of Cas9 protein transfection to the E. Derm cells. IFA was used to evaluate the expression of the Cas9 protein in E. Derm cells cultivated with EMEM + 10% FBS and Opti-MEM at 72 hpt. Cells were fixed and permeabilized, actin filaments were stained with Phalloidin (Red), the Cas9 protein was stained with Alexa Fluor 488 (green), and nuclei were stained with Hoechst 33342 (blue). Objective lens 20×, scale 50 µm (**A**). Graphical representation of the Cas9 expression level as the number of dots/cells following the transfection of E. Derm at 12, 24, 48, and 72 h post-transfection using the Opti-MEM and EMEM + 10%FBS. Three independent experiments were performed. Bars represent the mean ± standard deviation. Statistical analysis was performed by an unpaired *t*-test (**B**). Viability assessment of transfection protocol E. Derm cells cultured for 72 h in EMEM + 10% FBS or Opti-MEM. Three independent experiments were performed. Bars represent the mean ± standard deviation. Statistical analysis was performed by an unpaired *t*-test (*** *p* ≤ 0.001) (**C**).

**Figure 3 viruses-16-00409-f003:**
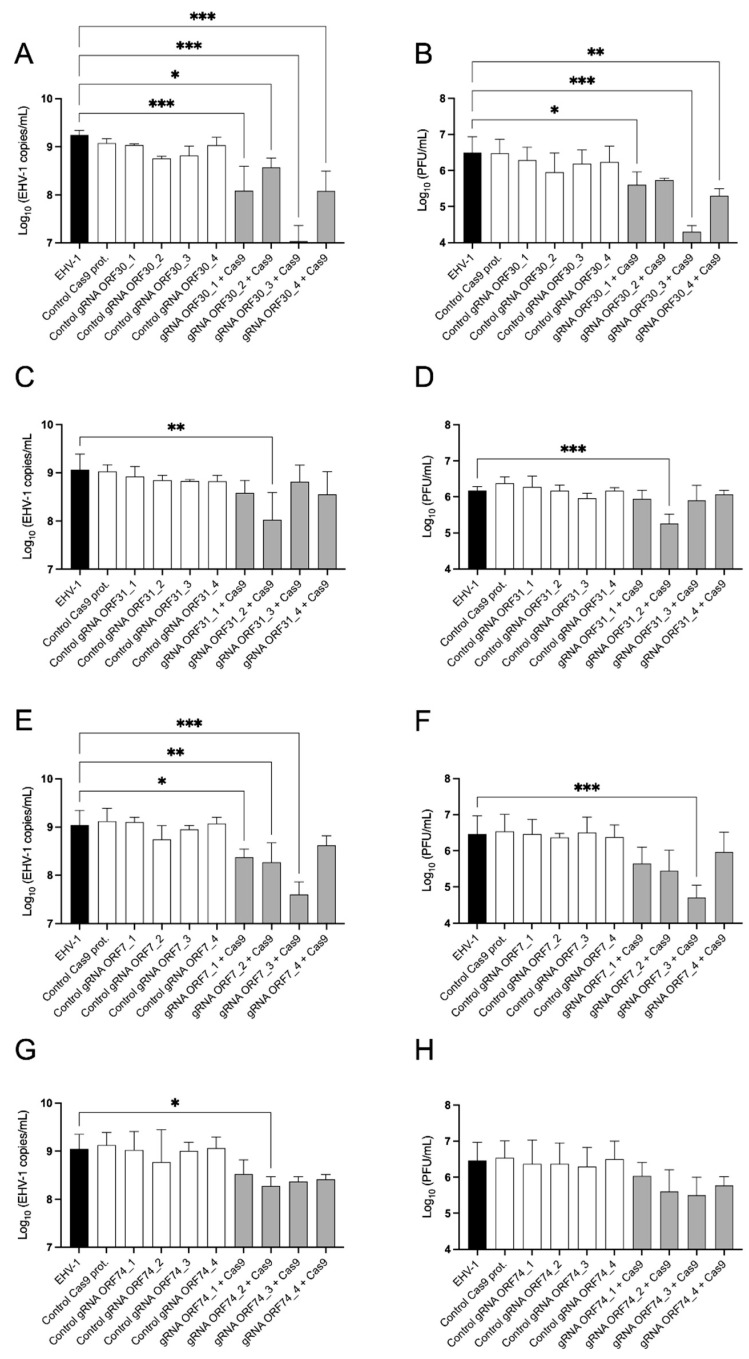
Effect of sgRNA-CRISPR/Cas9 on EHV-1 replication. E. Derm cells were infected with EHV-1 T953 at an MOI of 0.05 and subsequently transfected with the Cas9 protein and sgRNAs. Supernatants collected at 72 hpt were used for the viral copy number and viral titration determination. Histograms represent the EHV-1genomic copies/mL and PFU/mL of EHV-1 collected in the supernatant after treatments with sgRNAs targeting ORF30 (**A**,**B**), ORF31 (**C**,**D**), ORF7 (**E**,**F**), and ORF74 (**G**,**H**) genes. Three independent experiments were performed. Bars represent the mean ± standard deviation, and the significance was determined by One-way ANOVA with Dunnett’s post hoc test (*: *p* ≤ 0.05, **: *p* ≤ 0.01, ***: *p* ≤ 0.001).

**Figure 4 viruses-16-00409-f004:**
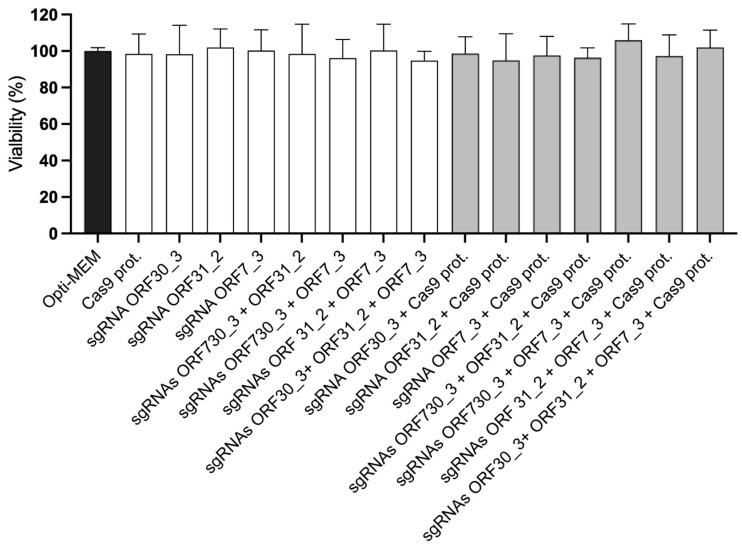
Viability of E. Derm cells in the presence or absence of Cas9 protein and specific sgRNAs. Cell viability was measured 72 hpt using the CellTiter Glo^®^ Luminescent Cell Viability Kit. Percentage of viability was reported to the control cells cultivated in Opti-MEM. Three independent experiments were performed in duplicate. Bars represent the mean ± standard deviation. Statistical analysis was performed by One-way ANOVA with Dunnett’s post hoc test.

**Figure 5 viruses-16-00409-f005:**
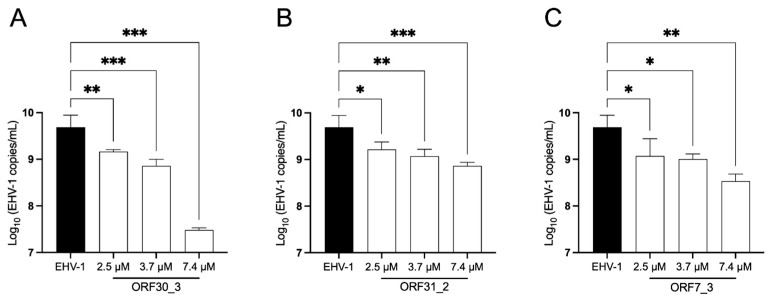
Dose-effect analysis of the selected sgRNAs targeting essential genes. E. Derm cells were infected with EHV-1 T953 strain at an MOI of 0.05 and successively transfected with Cas9 in the presence of different concentrations (2.5 µM, 3.7 µM, and 7.4 µM) of the selected sgRNAs: ORF30_3 (**A**), ORF31_2 (**B**), and ORF7_3 (**C**). Supernatants collected at 72 hpt were used for viral copy number determination. Three independent experiments were performed. Bars represent the mean ± standard deviation. Statistical analysis was performed by One-way ANOVA with Dunnett’s post hoc test (*: *p* ≤ 0.05, **: *p* ≤ 0.01, ***: *p* ≤ 0.001).

**Figure 6 viruses-16-00409-f006:**
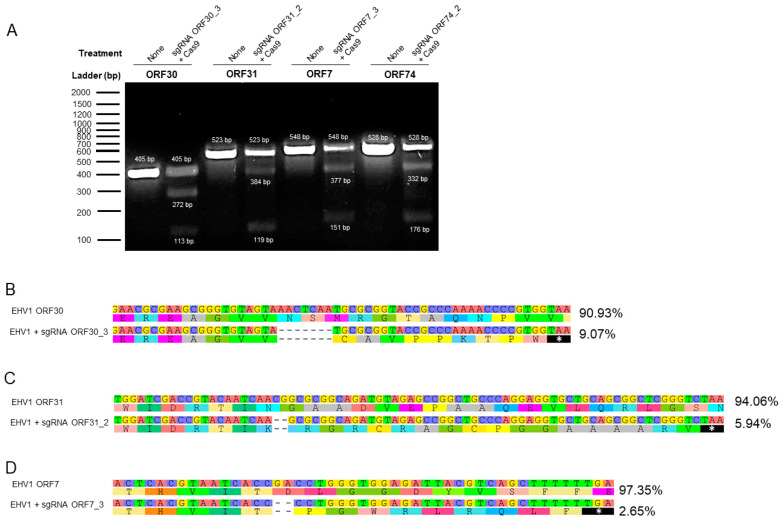
Site-specific cleavage detection after treatment with the selected sgRNAs and variant analysis. E. Derm cells were infected with EHV-1 T953 strain at an MOI of 0.05 and successively transfected with Cas9 in the presence of the selected sgRNAs ORF30_3, ORF31_2, and ORF7_3. Viral DNA was extracted from the cell pellet at 72 hpt, and DNA cleavage was detected using the GeneArt cleavage detection kit (**A**). A single band was observed from the non-treated infected cells, while three bands (one parental band and two cleaved bands as an indication of viral mutants) were observed from the infected cells treated with sgRNAs ORF30_3, ORF31_2, ORF7_3, and ORF74_2 in the presence of the Cas9 protein. The percentage of variant was analyzed by next-generation sequencing (NGS) for sgRNA ORF30_3 (**B**), sgRNA ORF31_2 (**C**), and sgRNA ORF7_3 (**D**). Sequencing of the PCR product indicates editing at the target sites. Mutant viruses with nucleotide deletions at target sites were identified. Stars on a black background indicate the STOP codon in the amino acid sequences (early termination) (NGS analysis was performed using Geneious software v.6.1.8).

**Figure 7 viruses-16-00409-f007:**
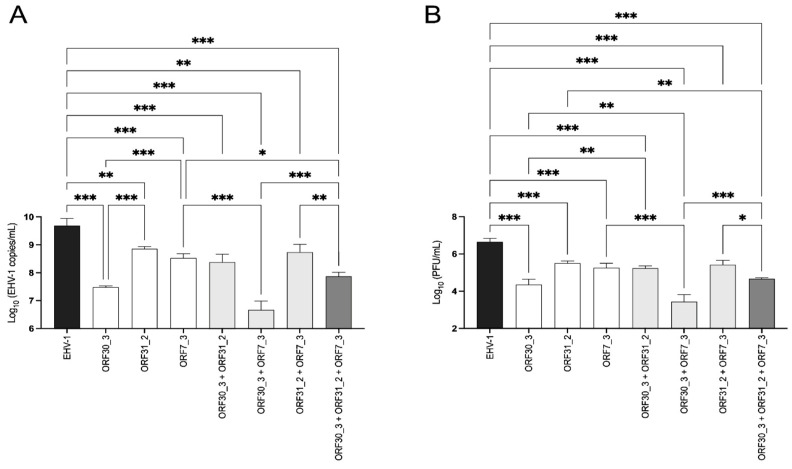
Effect of sgRNAs treatment on dual and triple combinations on EHV-1 replication. E. Derm cells were infected at an MOI of 0.05 and successively treated with different combinations of sgRNAs in the presence of the Cas9 protein. The supernatant was harvested at 72 hpt and used for viral copy number (**A**) and infectious particle titer (**B**) determination. Three independent experiments were performed. Bars represent the mean ± standard deviation. Statistical analysis was performed by One-way ANOVA with Tukey’ s post hoc test (*: *p* ≤ 0.05, **: *p* ≤ 0.01, ***: *p* ≤ 0.001).

**Table 1 viruses-16-00409-t001:** Sequences of sgRNAs used in this study.

EHV-1 Genes	sgRNA Names	Nucleotide Positions ^1^	Strand	Sequence (5′ to 3′)	PAM Sequence
**ORF30**	ORF30_01	95,418–95,437	−	TCGTTTGCATGCGTCCAGCG	CGG
	ORF30_02	54,122–54,141	+	AGGTGACGTGTCGCTCAACG	GGG
	ORF30_03	97,428–97,447	−	GGTGTAGTAAACTCAATGCG	CGG
	ORF30_04	95,959–95,978	−	TCGCCCGTATCACCCTAACG	CGG
**ORF31**	ORF31_01	56,100–56,119	+	TGGCTACGTACTGGGTCCGG	CGG
	ORF31_02	58,206–58,225	+	CGACCGTACAATCAACGGCG	CGG
	ORF31_03	57,321–57,340	+	GCGTGATTTCAAAATCCGCG	AGG
	ORF31_04	58,268–58,287	+	TCGGGTCTAACCCGGCCGCG	TGG
**ORF7**	ORF7_01	140,741–140,760	−	GGTAGTAAACAAGCGTACGA	GGG
	ORF7_02	8150–8169	+	AGCGCTCTAACGAGTTGAGG	GGG
	ORF7_03	141,740–141,759	−	ACTCACGTAATCACCGACCT	GGG
	ORF7_04	9864–9883	+	GTGTATTAGACGATAGCGGG	TGG
**ORF74**	ORF74_01	135,017–135,036	+	ATTGTACAACGGACATCCGG	AGG
	ORF74_02	135,539–135,558	+	TTGGTTCCGCGATACACCCG	AGG
	ORF74_03	14,651–14,670	−	CATCAGCGTACACGCGCGAG	TGG
	ORF74_04	135,353–135,372	+	AAGCAACGACCCCTCGGACG	AGG

PAM: protospacer adjacent motif; ORF: open reading frame; +: positive; −: negative. ^1^ Annotation based on the NCBI EHV-1 genome ID: KM593996.

## Data Availability

The data presented in this article are available in this published article and its Appendix A.

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
