# Peer review of "Modulation of Equid Herpesvirus-1 Replication Dynamics In Vitro Using CRISPR/Cas9-Assisted Genome Editing"

_viruses, 2024, doi:10.3390/v16030409_

Round 1

Reviewer 1 Report

Comments and Suggestions for Authors

The present work focus on a very interesting and powerful tool to modulate EHV-1 replication and infection in vitro: CRISPR/cas9 system. The results obtained are very encouraging to develop novel and efficacious strategy therapeutic and prophylactic approach against EHV-1.

The manuscript is very well written, clear, precise, and easy to understand. The authors clearly describe the materials and methods and the results. The limits of the study are also clearly discussed and results were compared with those obtained for other viruses.

Parts 2.2, 2.9, and 2.11: specify amplification program and concentrations of primers.

Figure 2A: specify the scale bar

Lines 410-414: which concentrations of sdRNAs were tested to evaluate the toxicity: 2.5, 3.7, and 7.4µM?

Figure 6B, C, D: specify the software used for NGS analysis in the caption.

Reviewer 2 Report

Comments and Suggestions for Authors

The authors present a very interesting and comprehensive study on the use of CRISPR-Cas9 as an antiviral for equine alphaherpesvirus. The experiments are all well described and presented data supports the conclusions drawn.

Minor suggestions:

Line 16 suggest revision “equid alphaherpesvirus-1 (EHV1)”

The abbreviation is used subsequently in the abstract but not introduced.

Line 56 suggest revision “the importance of this mutation for the”

I think this is singular as the mutation is one base in the gene sequence, the amino acid substitution is a consequence of that.

Line 365 Figure 2 The legend should state what the columns and error bars represent in Fig 2B and Fig 2C.

The same comment applies to subsequent figures with similar formats.

Line 588 suggest replacing “trial” with “experiment”.

Line 465 The stars on black backgrounds indicating stop codons are poorly visible in the figure. Please consider a different format to improve visibility.
